# A Novel Based Synthesis of Silver/Silver Chloride Nanoparticles from *Stachys emodi* Efficiently Controls *Erwinia carotovora*, the Causal Agent of Blackleg and Soft Rot of Potato

**DOI:** 10.3390/molecules28062500

**Published:** 2023-03-09

**Authors:** Shazia Dilbar, Hassan Sher, Dalal Nasser Binjawhar, Ahmad Ali, Iftikhar Ali

**Affiliations:** 1Centre for Plant Sciences and Biodiversity, University of Swat, Charbagh 19120, Pakistan; 2Department of Chemistry, College of Science, Princess Nourah Bint Abdulrahman University, P.O. Box 84428, Riyadh 11671, Saudi Arabia; 3Department of Genetics and Development, Columbia University Irving Medical Center, New York, NY 10032, USA

**Keywords:** silver/silver chloride nanoparticles, *Stachys emodi*, potato, antibacterial, *Erwinia carotovora*

## Abstract

In recent years, the biological synthesis of silver nanoparticles has captured researchers’ attention due to their unique chemical, physical and biological properties. In this study, we report an efficient, nonhazardous, and eco-friendly method for the production of antibacterial silver/silver chloride nanoparticles utilizing the leaf extract of *Stachys emodi*. The synthesis of se-Ag/AgClNPs was confirmed using UV-visible spectroscopy, DPPH free radical scavenging activity, Fourier transform infrared spectroscopy (FTIR), scanning electron microscopy (SEM), and X-ray diffraction (XRD). An intense peak absorbance was observed at 437 nm from the UV-visible analysis. The *Stachys emodi* extract showed the highest DPPH scavenging activity (89.4%). FTIR analysis detected various bands that indicated the presence of important functional groups. The SEM morphological study revealed spherical-shaped nanoparticles having a size ranging from 20 to 70 nm. The XRD pattern showed the formation of a spherical crystal of NPs. The antibacterial activity performed against *Erwinia carotovora* showed the maximum inhibition by centrifuged silver nanoparticles alone (se-Ag/AgClNPs) and in combination with leaf extract (se-Ag/AgClNPs + LE) and leaf extract (LE) of 98%, 93%, and 62% respectively. These findings suggested that biosynthesized NPs can be used to control plant pathogens effectively.

## 1. Introduction

Plant growth and development are affected by various biotic and abiotic factors affecting its yield and quality [1]. It is estimated that the world population is going to increase up to 10 billion by 2050 which will pressurize farmers for nutritious and safe food production in near future. The present food production systems are mainly threatened by diseases, pests, microorganisms, drought, and sudden climate changes [2].

Among these, phytopathogens are causing serious diseases in agricultural crops, resulting in global food insecurity [3]. The increased demand for vegetables and fruits such as potatoes, tomatoes, and eggplants has employed about 800 million individuals and contributes more than 33% of the world’s agricultural production [4]. The agricultural productivity of vegetables and fruits decreases due to the diseases caused by phytopathogens, which in turn increase the market price of such products [5]. Potato (*Solanum tuberosum* L.) is considered to be an attractive crop in the agricultural sector due to its high nutritional value as it is a good source of vitamins, proteins, energy, minerals, and carbohydrates [6]. Globally, potato is one of the most consumed foods, occupying fourth position after corn, rice, and wheat [7]. Potato is an important crop; it is not only utilized as a source of food, but as a feedstock for other industrial products. Potato crop yield is not only affected by abiotic stress such as high radiation, heat and cold stress, air pollutants (nitrogen dioxide and ozone), and drought stress, but also by biotic stress such as viral, fungal and bacterial diseases [8]. Potato crop is badly affected by various pathogenic diseases which lead to low production. The most prominent biotic factor which could affect the quantity and quality of potato are the bacterial diseases blackleg and soft rot caused by *Erwinia carotovora. Erwinia carotovora* is a Gram-negative bacterium with rod-shaped cells which decay most of the vegetables and fruits in the field and during shipment or its storage. These edibles are contaminated by the *E. carotovora* after rainfall and irrigation [9]. *E. carotovora* is one of the plant pathogens known to cause blackleg and soft rot diseases in potato crop [10]. A spreading maceration of the tuber tissue, frequently with a creamy consistency that darkens when exposed to air, is a sign of soft rot [11]. The disease is known as either potato blackleg disease or aerial stem rot when it affects the plant’s aerial components; its symptoms often start as a dark discoloration at the base of stem, at soil line and progress to the stems from infected tubers [12]. *E. carotovora* gain entry into the plant through some injuries and degrade its cell wall, followed by tissue maceration through pectolytic enzymes which lead to the soft rotting of the stems and fruits [13].

Globally, potato yield is affected annually by pests, weeds, and insects as well as diseases caused by different viruses, bacteria and fungi [14]. Viral, bacterial, fungal, and other plant diseases result in an annual loss of USD 1 billion worldwide [15].

Conventional methods are not useful enough to control these pathogens, while the frequent use of synthetic pesticides to manage plant diseases results in environmental pollution [16]. Several strategies are in use against phytopathogens for the control of various crop diseases for better crop production. The application of commercial pesticides has benefited farmers in one way by minimizing the effect of different crops’ ailments, but on the other hand, their frequent use leads to major upsets to human health as well as some plant-friendly, soil-borne microorganisms [17].

Nanotechnology deals with the manipulation and study of nanosized particles which can be used across various fields of science and technology [18]. Nanotechnology is of special concern in almost all fields of science and technology mainly because of the distinct chemical, biological, and physical characteristics of nanoparticles [19]. As a result of recent scientific advancements, metal nanoparticles are practical for producing drugs which can be used in various medical and industrial areas. Nanotechnology and nanoparticles have dominated the current generation which has attracted applications in daily life and technology to medicine, cosmetics, and space technology [20]. Nanobiotechnology is a modern field in which bulk materials are processed and converted into small particles having a size of 1–100 nm [21]. Nanobiotechnology has broad applications in the agricultural sector, e.g., to combat diseases and enhance crop productivity [22]. Recently, the use of nanobiotechnology in the management of plant diseases has gained substantial attention [23] due to the important role of nanomaterials in the control of plant pathogenic microorganisms and, thus, an improvement in crop productivity [24,25]. Nanobiotechnology possesses huge potential for the production of novel products which are advantageous to the environment as well as human health [26]. The green synthesis of nanoparticles by using medicinal plants has preference over physical and chemical methods as this method is eco-friendly, economic, and more effective [27]. Nanoparticles can be synthesized by using metals or nonmetals; however, metallic nanoparticles can be fabricated by using copper, cobalt, gold, silver, nickel etc. Metallic nanoparticles have received more attention because of their specific electrical, catalytic, and optical properties. Silver NPs are one of the main crucial and fabulous nanoparticles among several metallic nanoparticles which are involved in biomedical uses. Ag/AgClNPs play a noticeable role in nanoscience and nanotechnology, specifically in nanomedicine [28]. Among all nanoparticles, AgNPs hold a superior position due to their unique characters. Consequently, these nanoparticles have various applications including nanodevice fabrication, food technology, mechanics, biosensing, medicine, agricultural textiles, drug delivery, catalysis, electronics, and optics [29]. Because of the antiviral, antibacterial, and antifungal properties of nanoparticles, their use is increasing day by day in the agricultural field. In the past, the ancient Greeks used silver for wound healing, to treat ulcers, and as a preservative for food and water. Silver nanoparticles are used in different fields such as bone healing, bone cement, dental applications, and wound healing due to their antibacterial, antiviral, anticancer, and antifungal properties. Silver nanoparticles are applied as insecticides, pesticides, and growth promoters and against abiotic stresses [30]. The genus *Stachys* is considered to be a rich source of important plant secondary chemicals having therapeutic and commercial uses. The biological activities of *Stachys* are associated with the presence of different phytochemicals in various parts of the plant. In general, more than 200 compounds from the *Stachys* genus have been identified, and they fall into the following significant chemical groups: terpenes (e.g., diterpenes, iridoids, and triterpenes), essential oils, polyphenols (e.g., phenylethanoid glycosides, flavone derivatives, and lignans), and phenolic acids [31,32]. *Stachys* belongs to the third-largest genus of the Lamiaceae family with approximately 300 species, which are mostly perennial herbs and small shrubs mostly confined to the temperate regions in the Mediterranean, Asia, Southern Africa, and America. Many *Stachys* spp. have been used as traditional medicines for thousands of years. Most of the *Stachys* species are consumed to cure asthma, gastrointestinal diseases, the common cold, skin diseases, inflammation, and anxiety [33,34]. The *Stachys* species has been used by the traditional people of Europe, Japan, Iran, and in Chinese folk medicine as a tonic and for the treatment of other diseases [32,35,36]. *Stachys emodi,* commonly known as silky woundwort, belongs to the largest *Stachys* genus of the family Labiatae. The plant is a perennial herb with an erect stem reaching 60 cm tall, having 3–6 × 2–3 cm leaves, with multiflowered verticillasters in the axils of the leaves. The plant is distributed from Afghanistan and Pakistan (Kashmir) to Bhutan and NW India [37]. This study was designed to synthesize se-Ag/AgClNPs and evaluate them for various antibacterial activities against *E. carotovora*, the causal agent of blackleg and soft rot diseases in potato.

## 2. Results

### 2.1. DPPH Assay

The widely used method for determining free radical scavenging uses DPPH, a free radical with great stability. The *S. emodi* plant extracts showed strong antioxidant activity when they were assayed through DPPH free radical scavenging activity. The results showed that the *S. emodi* plant extract exhibited the highest DPPH scavenging activity (89.4) for the 1000 µg/mL concentration. Similarly, the obtained results of the plant sample concentration were compared with those of standard ascorbic acid (Figure 1).

### 2.2. Characterization of Silver/Silver Chloride Nanoparticles

The color of the solution started turning brown immediately after placing the solution in sunlight and turned completely dark brown after 20 min. This was due to the reduction of silver ion to silver/silver chloride nanoparticles in the reaction mixture [38]. The silver nanomaterial synthesis was achieved by using varying volumetric ratios (1:9, 2:8, 3:7, 4:6, 5:5, 6:4, 7:3, 8:2, and 9:1 *v*/*v*) of the *S. emodi* extract and silver nitrate solution. The UV-visible spectrum of the reaction mixture was recorded after 24 h. Nine various peaks were obtained for different ratios. The maximum absorbance was observed at 437 nm (Figure 2) and pH 11 for the 6:4 (*v*/*v*) ratio, where its pointed peak indicated the formation of spherical-shape (se-Ag/AgClNPs) silver/silver chloride nanoparticles [39].

The FTIR pattern was used to study the various functional groups which might be involved in se-Ag/AgClNP synthesis and could play an important role as a stabilizing agent. The FTIR spectral analysis showed various peaks for different functional groups. A small broad peak at 2035 cm^−1^ was observed due to C=C=N stretching, which could be a ketenimine-like compound. A small peak at 1975 cm^−1^ was observed due to C-H bending of the possible aromatic compound. A broad peak was observed at 1576 cm^−1^ for the N-H bending of the amine compound (Figure 3).

Scanning electron microscopic analysis was used to verify the morphology and size of the synthesized silver/silver chloride nanoparticles. The obtained SEM results showed that the produced nanoparticles had random morphology with spherical-shaped structures detected in the micrograph. The obtained results showed the size of the synthesized nanoparticles was in the range of 20 to 70 nm (Figure 4).

The crystalline nature of the sliver/silver chloride nanoparticles was confirmed by using XRD analysis at a 2θ angle ranging from 10° to 80°. The XRD diffraction peaks situated at 38.10°, 44.1°, 64.41°, and 77.35° are indexed to the (111), (200), (220), and (311) crystalline planes of pure Ag nanoparticles with a face-centered cubic structure according to the reference database in the Joint Committee on Powder Diffraction Standards (JCPDS) library (JCPDS, file No. 04-0783). The other dominant and clear five peaks at 27.71°, 32.14°, 46.11°, 54.73°, and 57.40° are attributed to planes (210), (122), (231), (142), and (241) of the cubic phase of silver chloride (AgCl) crystal (JCPDS No. 31-1238). The XRD results show that the biosynthesized Ag/AgClNPs were in the shape of spherical crystals (Figure 5). The Debye–Scherrer equation measured the average crystallite size as 38 nm for the bio-fabricated se-Ag/AgClNPs which validates SEM results.

### 2.3. Antibacterial Activity

The antibacterial activity against *E. carotovora* resulted in significant inhibition by various concentrations (500 µg mL^−1^, 250 µg mL^−1^, 100 µg mL^−1^, 80 µg mL^−1^, 50 µg mL^−1^, 20 µg mL^−1^, and 10 µg mL^−1^). The centrifuged nanoparticles in combination with the leaf extract (se-Ag/AgClNPs + LE) at a concentration of 500 µg mL^−1^ showed a maximum inhibition of 98%. The centrifuged nanoparticles alone (se-Ag/AgClNPs) inhibited the growth of the bacteria by 93%, while the leaf extract alone (LE) showed an optimal inhibition of 62%. The control treatment showed no inhibition of the cell growth of *E. carotovora*. The inhibition patterns of the various concentrations of se-Ag/AgClNPs + PE, se-Ag/AgClNPs, and PE are shown in Figure 6. 

## 3. Discussion

Plants produce different types of secondary metabolites which are potentially active against various insects and phytopathogens. As compared to commercial fungicides and pesticides, medicinal plants have more antifungal and antibacterial properties due to the presence of secondary metabolites which are more active in controlling plant diseases and are eco-friendly with fewer side effects [40]. As compared to commercial fungicides and pesticides, medicinal plants have more antifungal and antibacterial properties due to the presence of secondary metabolites which are more active in controlling plant diseases and are eco-friendly with fewer side effects [41]. The spectrophotometer peak is dependent on the size of the nanoparticles. A smaller particle size represents peaks at a shorter wavelength while a larger particle size indicates a longer wavelength peak [42]. Our findings regarding UV-visible analysis were in compliance with those of previously described studies [43] in which silver/silver chloride nanoparticle peaks were observed at around 420 nm. Similar results were also reported by Patra et al. [44] using *Pisum sativum* plant extract and Kup et al. [45] using a plant extract of *Aesculus hippocastanum*. They used various techniques to characterize their synthesized nanomaterials. According to the UV-visible analysis, the formation of Ag-NPs was observed at a wavelength above 420 nm.

The color change in the mixture from violet blue to yellow proved the reduction of DPPH radical by the antioxidant compounds in plants [46]. This is due to the potential of methanolic extract in *S. emodi* plants as antioxidants. The highest DPPH free radical scavenging activity was shown by the plant extract at a concentration of 1000 µg/mL, which was 89.4%. Similar results were shown by the previous studies by Tatarczak et al. [47] where the DPPH radical was reduced by the phytochemicals in plants, proving its strong antioxidant activity. Khan et al. [48] synthesized Au/MgO nanomaterial by using *Tagetes minuta* which exhibit excellent antioxidant activity with 82% scavenging capability.

FT-IR revealed that stretching in the band from 3000 to 2000 cm^−1^ revealed good bonding between the functional groups and the Ag. The observed FTIR spectrum of the synthesized nanoparticles was in complete agreement with previous studies [49]. The FTIR pattern showed the presence of biological groups in the *S. emodi* extracts which could be involved in reducing and capping the biosynthesized nanoparticles (se-Ag/AgClNPs). The agents which could be responsible for the bioreduction and stabilization of silver ions into silver NPs present in *S. emodi* extract were confirmed by the FTIR pattern. The obtained bands of FTIR could be attributed mainly to the phenols, terpenoids, and flavonoids present in *S. emodi* plant extracts. The present study agreed with the study by Mohamed et al. [50] which also suggested that flavonoids, phenols, and proteins could be the reducing and stabilizing agents of silver/silver chloride nanoparticles.

Our SEM observation of the synthesized nanoparticles was in complete accordance with that previously observed by Khan et al. [51], who prepared silver nanoparticles by using *Mentha spicata.* Their SEM results at different magnifications showed spherical-shaped particles with size ranges from 21 to 82 nm. According to Yousaf et al. [52], silver nanoparticles were synthesized from the extracts of *Achillea millefolium* L. Their SEM results had an average diameter of 14.27, 18.49, and 20.77 nm with spherical, cubical, and rectangular morphology which positively correlates with the present study. This study suggested that the obtained se-Ag/AgClNPs were capped by biomolecules present in the *S.emodi* plant extracts and these metabolites may be manipulated by metallic silver to biogenically synthesize silver/silver chloride nanoparticles. The *S. emodi* NPs’ size could also be detected from the sharpness and broadness of the XRD peaks. The Figure 5 peaks show that se-Ag/AgClNPs were in the nanosize range. Our XRD results were generally in accordance with those XRD patterns previously described by Hashemi et al. [53] which had the same peaks. Our results are in positive agreement with Sing et al. [54]; their XRD peaks were very strong and revealed that the synthesized Ag/AgClNPs were in the nanosize range and had a crystalline nature.

Plants belonging to the *Stachys* genus are very medicinal and have been used since early eras as traditional medicine to cure many problems such as gout, cough, fever, asthma, earaches, genital tumor, abdominal cramps, menstrual disorder, and dizziness. Advanced research shows that *Stachys* genus plant extracts have strong antifungal, antibacterial, antinephritic, antioxidant, and anti-inflammatory activities [31]. Previously, Shakeri et al. [55] reported the strong antibacterial efficacy of *Stachys* against *Staphylococcus aureus*, *Bacillus cereus*, *Staphylococcus epidermidis*, *Escherichia coli*, *Salmonella typhi,* and *Pseudomonas aeruginosa.* Jan et al. [56] also observed the antibacterial effects of *Stachys* against various bacteria. They showed that ethyl acetate, aqueous, n-hexane, and ethanolic extracts of the *Stachys parviflora* plant showed strong antimicrobial activity against six bacteria (*Bacillus atrophaeus*, *Staphylococcus aureus*, *Pseudomonas aeruginosa*, *Salmonella typhi*, *Escherichia coli,* and *Bacillus subtilis*) and one fungi (*Candida albicans).*

The antibacterial activity of the study showed similar results to those previously reported studies that suggested that biosynthesized Ag/AgClNPs (by using grape pomace aqueous extract) have potential in controlling the growth of *E. carotovora* [57]. *E. carotovora* is a Gram-negative bacterium which affects most vegetables and fruits in the field and during shipment or in storage [58]. *E. carotovora* infect the host plant through some injuries and degrade its cell wall, followed by tissue maceration leading to the soft rotting of stems and fruits [9,59].

In the current study, the antibacterial effect of the *S. emodi* leaf extracts (500 µg/mL–10 µg/mL) was tested against *E. carotovora*. High concentrations (500 µg/mL and 250 µg/mL) showed promising results against the bacteria while the lowest activity was observed for the lowest (10 µg/mL) concentration of the leaf extract. Similarly, high concentrations (500 µg /mL, 250 µg/mL, and 100 µg/mL) of plant-coated silver/silver chloride nanoparticles (se-Ag/AgClNPs + PE) had high antibacterial activity as compared to the lowest concentrations (80 µg/mL to 10 µg/mL). The activity of the centrifuged nanoparticles (se-Ag/AgClNPs) was high for the 500 µg/mL and 250 µg/mL concentrations. Overall, the activity of the plant-coated nanoparticles was superior to the centrifuged nanoparticles and plant extracts alone. This might be due to the synergism of secondary metabolites with silver ions which makes its activity more efficient. The antibacterial activities of the present study agreed with those of Arif et al.’s study [38], in which silver nanoparticles synthesized from *Euphorbia wallichii* were tested against phytopathogens. Our study is in positive correlation with the previously reported study by Balachandar et al. [60] who studied the activities of biologically synthesized nanoparticles against various phytopathogens, and noticed a strong growth inhibition of the plant pathogens. NPs synthesized by using *Eucalyptus camaldulensis* were tested against various bacteria and were found to significantly reduce the Gram-negative bacteria growth. The antibacterial character of Ag/AgClNPs prepared from *E. camaldulensis* could be ascribed to the small particle size and high surface-to-volume ratio, which let the nanoparticles interact with bacterial membranes [61]. According to a proposed mechanism which describes how silver particles act, due to their small size and spherical shape, they can penetrate bacterial cell walls and can increase their permeability by bringing some structural changes; these changes include the generation of pores in the bacterial cell wall through reactive oxygen species production. Silver ions can also damage important cell enzymes, proteins, and nucleic acids of the bacteria, resulting in bacterial cell death [62].

## 4. Materials and Methods

### 4.1. Preparation of Leaf Extract

Healthy plant specimens were collected, washed thoroughly, and dried up at room temperature. The dried specimens were ground into a fine powder and used for the synthesis of silver/silver chloride nanoparticles. For the preparation of leaf extract, 1 g of the ground powder was mixed in 100 mL of distilled water and the solution was heated at 40–50 °C for 15 min on a hot plate. The solution was left to cool down and then it was filtered with the help of Whatman No. 1 filter paper (pore size of 11 µm) and was stored in a refrigerator at 4 °C for further use.

### 4.2. Antioxidant Activity

The DPPH free radical scavenging capacity of the sample plant was determined according to the protocol of Govindappa et al. [63] with a minor modification. The plant solution was prepared by taking 10 mg of the powdered plant in 10 mL of methanol. Through the two-fold dilution of the plant stock solution, 5 different concentrations (1000 µg/mL, 500 µg/mL, 250 µg/mL, 125 µg/mL, and 62.5 µg/mL) were formed. The DPPH solution was already prepared and stored at room temperature in the dark. A total of 1 mL of the DPPH solution was added to 2 mL of these diluted samples of the plant extract and kept for incubation in the dark for 30 min. The absorbance of all concentrations was measured with a Multiskan TM Sky Microplate Spectrophotometer (MAN0018930, Santa Clara, CA, USA) at 517 nm while ascorbic acid was used as standard. The percent activity was calculated with the given formula.
% antioxidant activity = (OD of the control − OD of the sample × 100)/control OD(1)

### 4.3. Biosynthesis of Silver/Silver Chloride Nanoparticles

Following the procedures of Arif et al. [63] and Ul Haq et al. [64] with certain modifications, the green synthesis of nanoparticles was accomplished. The diluted leaf extract (2.5 mg/mL) solution was mixed appropriately with silver nitrate (4 mM) solution at equal volume (1:1) and was kept under sunlight for 20 min. The mixture was adjusted at different pHs ranging from 6 to 12. For the separation of the synthesized Ag/AgClNPs, the solution was centrifuged (Centrifuge 5425, Eppendorf, Hamburg, Germany) at 15,000 rpm for 15 min. The residual settled material was collected in a distinct tube and was then dissolved in deionized water followed by centrifugation again at 12,000 rpm for 10 min. This was repeated multiple times and the obtained pure nanoparticles were subjected to various characterizations.

### 4.4. Characterization of Synthesized Particles

The biosynthesis of se-Ag/AgClNPs was confirmed through various characterization techniques, which were as follows:

Fourier transform infrared spectrophotometric analysis was carried out using an FTIR (Spectrum two-103385; Waltham, MA, USA) spectrophotometer equipped with ATR. The FTIR spectroscopic analysis was performed between the ranges of 4000 and 400 cm^−1^. The various functional groups were identified by comparing the observed peaks with an IR spectrum table.

Scanning electron microscopy (JSM-5910, JEOL, Tokyo, Japan) was used to find out the morphology and distribution of the nanoparticles.

The X-ray diffraction (XRD) analysis of the silver/silver chloride nanoparticles was carried out using an X-ray diffractometer (Model: X-3532, JEOL, Tokyo, Japan). The XRD patterns were evaluated to find out the peak intensity, position, and width. The mean crystallite size was measured using Debye–Scherrer’s formula.

### 4.5. Antibacterial Activity

The antibacterial activity of the green synthesized se-Ag/AgClNPs against *E. carotovora* was accomplished using the methods of Ahmad et al. [65] with certain modifications. The freshly grown culture of *E. carotovora* was acquired from (FCBP-PB-421) First Fungal Culture Bank of Pakistan (FCBP), Institute of Agricultural Sciences (IAGS) University of Punjab, Lahore, Pakistan and inoculated in nutrient broth and placed overnight at 28 °C in an incubator (FTC-90E Velp Scientifica, Lombardia, Italy). The activity was implemented with a microtiter plate (96-well) assay with various concentrations (500 µg mL^−1^, 250 µg mL^−1^, 100 µg mL^−1^, 80 µg mL^−1^, 50 µg mL^−1^, 20 µg mL^−1^, and 10 µg mL^−1^) of centrifuged silver nanoparticles alone (se-Ag/AgClNPs) and in combination with leaf extract (se-Ag/AgClNPs + LE) and leaf extract (LE) alone. 150 µL concentration of each treatment and 150 µL of *E. carotovora* suspension were poured into each well of the microtiter plate. The control well was adjusted with bacterial suspension. The optical density (OD) at 600 nm was recorded immediately and placed in a shaking incubator for 24 h. After 24 h, the OD was again read at 600 nm and the bacterial growth inhibition was calculated using the given formula:Bacterial growth inhibition = Control − Treatment/Control × 100(2)

## 5. Conclusions

In the present study, we showed the efficient biosynthesis of silver/silver chloride nanoparticles and their antibacterial screening against *E. carotovora* using *S. emodi*. The characterizations of the prepared nanoparticles showed a significant biosynthesis of stable silver/silver chloride nanoparticles. Our study showed that the size of the nanoparticles ranged from 20 to 70 nm with an average diameter of 38 nm. Moreover, the antibacterial activity resulted in the significant growth inhibition of *E. carotovora* by the biogenically synthesized silver/silver chloride nanoparticles. The study concluded that biosynthesized Ag/AgClNPs have the potential to control the growth of *E. carotovora* through in vitro activities. It is recommended to evaluate the potential of se-Ag/AgClNPs through in planta means. However, further studies should confirm the effectiveness of these nanoparticles against other plant pathogens to protect important crops.

## Figures and Tables

**Figure 1 molecules-28-02500-f001:**
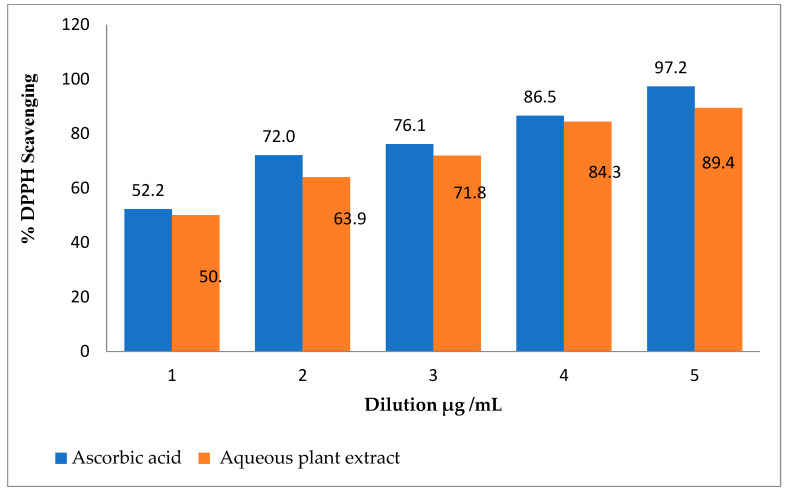
Antioxidant activity of *S. emodi* extract against DPPH radicals.

**Figure 2 molecules-28-02500-f002:**
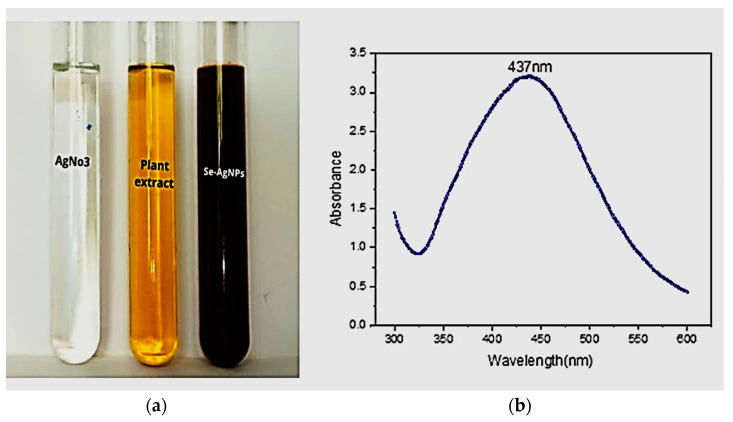
Color change to dark brown after exposing mixture to sunlight for 20 min (**a**), and UV-visible spectrum post-24 h of the reaction (**b**).

**Figure 3 molecules-28-02500-f003:**
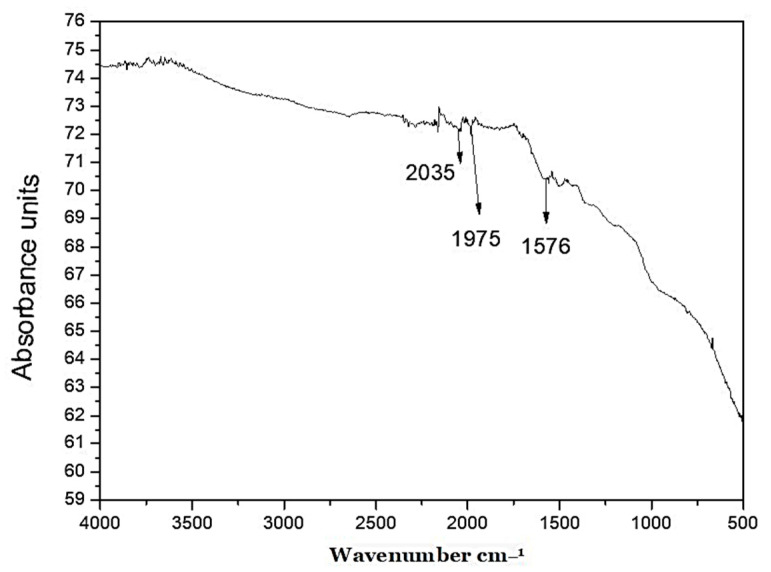
FTIR analysis of the biosynthesized silver/silver chloride nanoparticles.

**Figure 4 molecules-28-02500-f004:**
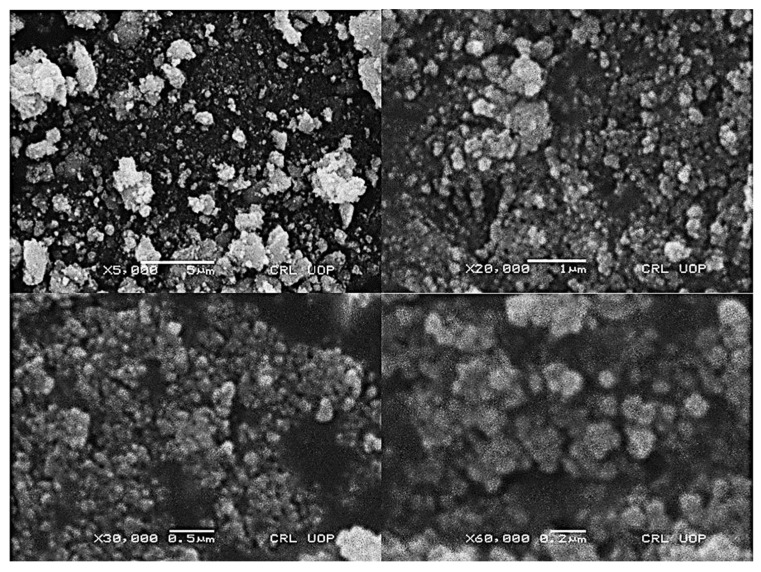
Scanning electron microscopic analysis of the *S. emodi*-mediated silver/silver chloride nanoparticles.

**Figure 5 molecules-28-02500-f005:**
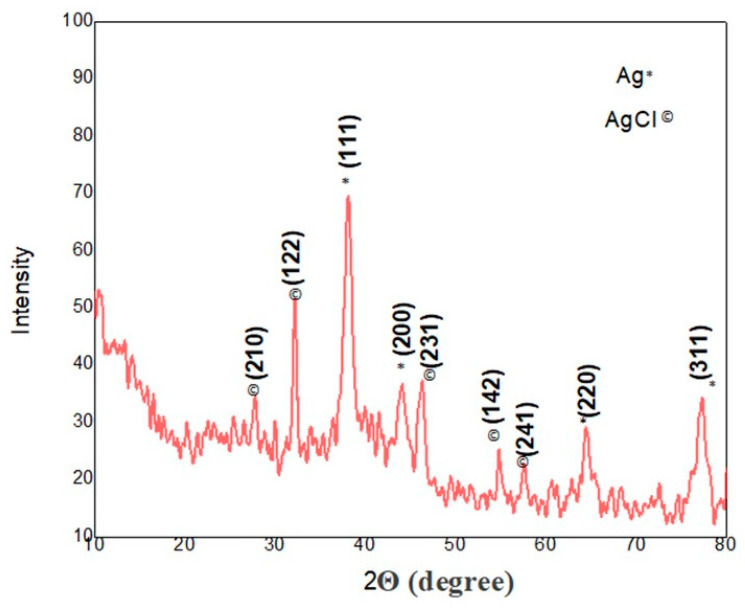
XRD analysis showing crystalline planes for the synthesized silver/silver chloride nanoparticles.

**Figure 6 molecules-28-02500-f006:**
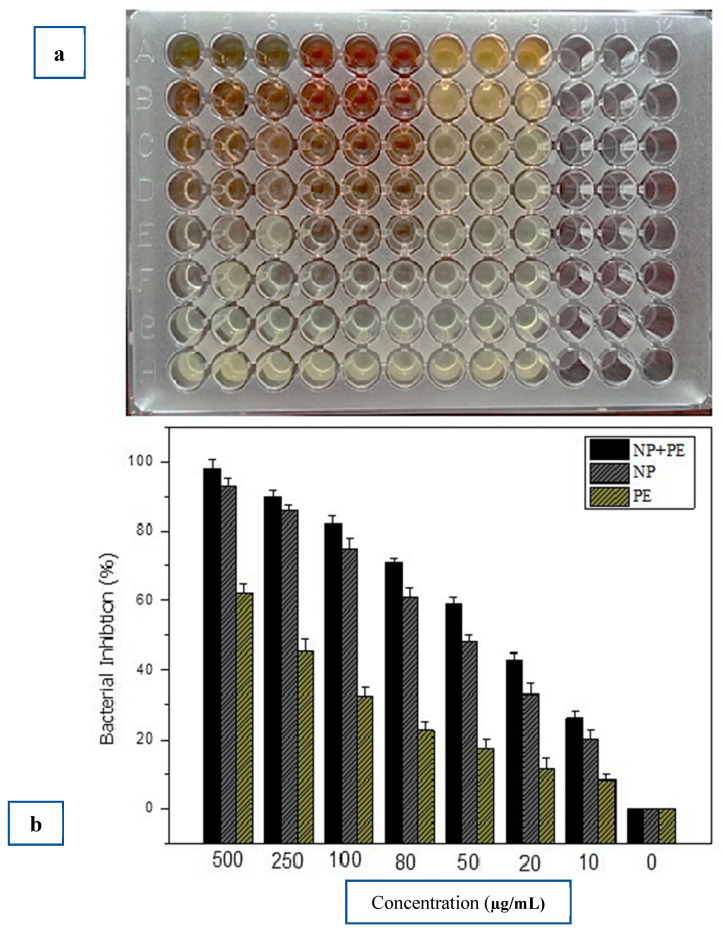
(**a**). Microtiterplate loaded with centrifuged NPs, plant extract and NPs in combination with plant extract along with bacteria. (**b**). Antibacterial activity graph showing inhibition of *E. corotovora* by various concentrations of silver/silver chloride nanoparticles alone (se-Ag/AgClNPs), in combination with leaf extract (se-Ag/AgClNPs + LE), and leaf extract alone (LE).

## Data Availability

All authors confirm that the generated data are available in the manuscript.

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
