# Peer review of "A Novel Based Synthesis of Silver/Silver Chloride Nanoparticles from Stachys emodi Efficiently Controls Erwinia carotovora, the Causal Agent of Blackleg and Soft Rot of Potato"

_molecules, 2023, doi:10.3390/molecules28062500_

Round 1

Reviewer 1 Report (Previous Reviewer 3)

The authors have modified the manuscript according to my previous comments

References in text should include the author's name before the reference number , as is indicated in Citations Style Guide for MDPI Journals. As exemple: to the protocol of [38] with minor modification... should be-----  to the protocol of Govindappa, et al. [38] with minor modificationt.  Correct this type of cites in the manuscript 

Author Response

Reviewer 1

Comment 1: References in text should include the author's name before the reference number, as is indicated in Citations Style Guide for MDPI Journals. As exemple: to the protocol of [38] with minor modification... should be to the protocol of Govindappa, et al. [38] with minor modificationt.  Correct this type of cites in the manuscript.

Response: thank you for your valuable comment. The correction has been done according to the mentioned style as well as been highlighted in the manuscript.

Reviewer 2 Report (Previous Reviewer 2)

Dear author,

Some improvements but Your article still contains severas flaws; 

1- abstract : why do you use “”  to mention techniques? 
2- go through the entire manuscript to check against typos and grammatical mistakes, punctuation and spacing !

3- follow the authors guidelines correctly! when you cite materials then you need to mention not only model and manufacturer but also city, state and country; also you new to specify if any conflicts of interest, acknowledgements, ERB approval number, funding number of any etc.

4- the references must be accurate, recent > 50% and uniformly edited according to the journal’s requirements 

5- statistical power p-value (*, **, ***  etc) must be mentioned within the figures and tables; statistical differences must be shown.

best,

the reviewer 

Author Response

Reviewer 2

Some improvements but Your article still contains several flaws; 

Comment 1- abstract: why do you use “” to mention techniques? 

Response: Thank you for valuable comment. “” have been removed.

Comment 2- go through the entire manuscript to check against typos and grammatical mistakes, punctuation and spacing!

Response: Thank you for valuable comments. The language has been revised and manuscript has adjusted according to your precious suggestions.

Comment 3- follow the authors guidelines correctly! when you cite materials then you need to mention not only model and manufacturer but also city, state and country; also you new to specify if any conflicts of interest, acknowledgements, ERB approval number, funding number of any etc.

Response: Thank you for valuable comments. City, state and country has been mentioned with all the instruments. Further conflicts of interest, acknowledgements, and funding number are also mentioned.

Comment 4- the references must be accurate, recent > 50% and uniformly edited according to the journal’s requirements 

Response: Thank you for valuable comments. The references have been corrected as per the suggestion.

Comment 5- statistical power p-value (*, **, ***  etc) must be mentioned within the figures and tables; statistical differences must be shown.

Response: Thanks for your positive comment. We checked and added statistical differences and p- values when applicable.

Reviewer 3 Report (Previous Reviewer 1)

Again, the introduction section should be well structured, i.e., it should connect the importance of the work in a gradual manner. Thus, the authors are advised to improve and rewrite some portion of the introduction. 

Novelty/ aim of the study. The authors should focus on the exacerbation of the current research problem, the existing challenges, and the available solutions

Materials and Methods. The material synthesis section is written too general as a kind of report, and not suitable for a scientific paper. It should be rewritten in a concise and concrete form including deep discussion. Add the complete instrumental details used in this current study according to the journal’s guidelines.

Reasults/ Discussion must be considerably improved/ more technically presented.. Again, please improve the quality of ALL figures; please uniformize the size of figure, font and size of descriptors, etc.

When refer to previous reported studies (example: Similar results were shown by the previous studies [50, 51]), please use significant values.

The authors mustcarefully revise language of the manuscript before publication and the whole article should be adjusted based on journal style.

Finally, I consider that the paper is still not proper for publication in the present format and must be "Reject".

 Author Response

Reviewer 3

Comment 1:  the introduction section should be well structured, i.e., it should connect the importance of the work in a gradual manner. Thus, the authors are advised to improve and rewrite some portion of the introduction. 

Response: thank you for your valuable comments. The amendment has been done as per suggestion.

Comment 2: Novelty/ aim of the study. The authors should focus on the exacerbation of the current research problem, the existing challenges, and the available solutions.

Response: Thank you for valuable comments. Based on our knowledge, there is no published report on the biogenic synthesis of silver nanoparticles through green chemistry by using Stachys emodi plant extracts as reducing and stabilizing agent. So the present study was aim to synthesized Se-AgNPs and evaluate it for various antibacterial activity against E. carotovora the causal agent of blackleg and soft rot diseases in potato.

Comment 3: Materials and Methods. The material synthesis section is written too general as a kind of report, and not suitable for a scientific paper. It should be rewritten in a concise and concrete form including deep discussion. Add the complete instrumental details used in this current study according to the journal’s guidelines.

Response: Thank you for valuable comments.  Added your precious comments in the manuscript.

Comment 4: Results/ Discussion must be considerably improved/ more technically presented. Again, please improve the quality of ALL figures; please uniformed the size of figure, font and size of descriptors, etc. When refer to previous reported studies (example: Similar results were shown by the previous studies [50, 51]), please use significant values.

Response:  Thank you for valuable comments.  The quality of figures has been improved and size of figures has been uniformed as well as the font and size of description. The discussion has been changed as suggested and has been highlighted in the manuscript.

Comment 5: The authors must carefully revise language of the manuscript before publication and the whole article should be adjusted based on journal style.

Response: Thank you for valuable comments.  The language has been revised and manuscript has been adjusted at journal style.

Comment 6: Finally, I consider that the paper is still not proper for publication in the present format and must be "Reject".

Response:  We have tried best to incorporate all valuable suggestions of the reviewer.

Round 2

Reviewer 3 Report (Previous Reviewer 1)

Based on this revised version of the manuscript, unfortunately there are still unanswered questions and weaknesses in this paper. Please revise it accordingly.

Author Response

Dear Editors,

Thank you for giving us the opportunity to submit a revised draft of our manuscript “A novel based synthesis of silver nanoparticles from Stachys emodi efficiently controls Erwinia carotovora; the causal agent of blackleg and soft rot of Potato. We appreciate the time and effort that you and the reviewers have dedicated to providing your valuable feedback on our manuscript. We are grateful to the reviewer for their insightful comments on our paper. We have been able to incorporate changes to reflect most of the suggestions provided by the reviewers. All the changes have been highlighted within the manuscript.

Reviewer 3

Comment 1:  the introduction section should be well structured, i.e., it should connect the importance of the work in a gradual manner. Thus, the authors are advised to improve and rewrite some portion of the introduction. 

Response: thank you for your valuable comments. In Introduction section, first importance of potato crop has been discussed with its nutritional value, its position among other vegetables and its annual production. Then discussed the main diseases and its causal agents specifically Erwinia carotovora.  It is also being discussed about the bacteria and the mechanism through which it gets entered the host pant and caused disease. The control measures of the disease are also mentioned in the introduction part, then discussed about the most advance method i.e Nanotechnology, Nanoparticles through green chemistry. It also been described in the introduction section about the plant genus and specie in which silver nanoparticles has been synthesized. This study was designed to synthesize se-AgNPs and evaluates it for various antibacterial activities against E. carotovora, the causal agent of blackleg and soft rot diseases in potato.

Comment 2: Novelty/ aim of the study. The authors should focus on the exacerbation of the current research problem, the existing challenges, and the available solutions.

Response: Thank you for valuable comments. Based on our knowledge, there is no published report on the biogenic synthesis of silver nanoparticles through green chemistry by using Stachys emodi plant extracts as reducing and stabilizing agent. So the present study was aim to synthesized Se-AgNPs and evaluate it for various antibacterial activity against E. carotovora the causal agent of blackleg and soft rot diseases in potato.

Comment 3: Materials and Methods. The material synthesis section is written too general as a kind of report, and not suitable for a scientific paper. It should be rewritten in a concise and concrete form including deep discussion. Add the complete instrumental details used in this current study according to the journal’s guidelines.

Response: Thank you for valuable comments.  Added your precious comments in the manuscript.

Comment 4: Results/ Discussion must be considerably improved/ more technically presented. Again, please improve the quality of ALL figures; please uniformed the size of figure, font and size of descriptors, etc. When refer to previous reported studies (example: Similar results were shown by the previous studies [50, 51]), please use significant values.

Response:  Thank you for valuable comments.  The quality of figures has been improved and size of figures has been uniformed as well as the font and size of description. The discussion has been changed as suggested by the reviewer and has been highlighted in the manuscript.

Comment 5: The authors must carefully revise language of the manuscript before publication and the whole article should be adjusted based on journal style.

Response: Thank you for valuable comments.  The language has been revised and manuscript has been adjusted at journal style.

Comment 6: Finally, I consider that the paper is still not proper for publication in the present format and must be "Reject".

Response:  We have tried best to incorporate all valuable suggestions of the reviewer.

Round 3

Reviewer 3 Report (Previous Reviewer 1)

Dear authors, finally you did a good job making quality revision. In this regard, I will recommend your manuscript for publication.

Author Response

Academic Editor Notes: Unfortunately, the XRD section is still not OK. In the present version, the authors just indexed the crystallographic planes, but only the ones at 38.1, 44.2, 64.4 and 77.4 belong to the elemental silver. Maybe the authors found the information that all planes belong to the elemental Ag in literature, but they should check more papers, and most importantly to check it directly from the database (I did just that). Other peaks belong to other crystallographic phase. This is also clear from the shape of slightly different shape of the other peaks. The auhtors should clearly identify what is that and modify the results text accordingly.

Dear Editor, thanks a lot for correction in the crystallographic planes identification. We read some more relevant publications like (Zhao, X.; Zhang, J.; Wang, B.; Zada, A.; Humayun, M. Biochemical Synthesis of Ag/AgCl Nanoparticles for Visible-Light-Driven Photocatalytic Removal of Colored Dyes. Materials 2015, 8, 2043-2053. https://doi.org/10.3390/ma8052043) and also checked through database and found that characteristic peaks at 27.7, 32.1, 46.1, 54.7 and 57.4 are AgCl. We also modified the results text accordingly with track changes.

This manuscript is a resubmission of an earlier submission. The following is a list of the peer review reports and author responses from that submission.

Round 1

Reviewer 1 Report

The manuscript must follow the journal guidelines, please check and revise it accordingly.

Abstract. This section has been presented in general form and contains many information that can be omitted or reduced.

The introduction section should be well structured, i.e., it should connect the importance of the work in a gradual manner. Thus, the authors are advised to improve and rewrite some portion of the introduction. 

Novelty/ aim of the study.  The authors should focus on the exacerbation of the current research problem, the existing challenges, and the available solutions

Materials and Methods. The material synthesis section is written too general as a kind of report, and not suitable for a scientific paper. It should be rewritten in a concise and concrete form including deep discussion. Add the complete instrumental details used in this current study.

Reasults/ Discussion. Please improve the quality of Figures; please uniformize the size of figure, font and size of descriptors, etc.

Sentences as Our studies are in positive correlation with that of previous reported studies [73,74,75,76 ]”are confusing. Please use significant values.

References. Too many references (74 in this case) may question on novelty of the research. Moreover, the self-cites must only ever be used where they are genuinely needed and relevant for the articles in which they are included. Please revise.

 The authors must revise language of the manuscript before publication and the whole article should be adjusted based on journal style.

Finally, I consider that the paper is not proper for publication in the present format and must be "Reject". Nevertheless, the efforts of performing all the experiments have been significant and I hope that in the near future all the issues will be solved.

Reviewer 2 Report

Dear authors,

your manuscript fits the scope of the journal. Your work showed the efficient biosynthesis of silver nanoparticles and its antibacterial screening against E. carotovora using S. emodi.  It is fairly presented structured; the conclusions are at least partially supported by the data.
however several major points needs to be considered prior to publication:

1- the data need to be more deeply analyzed and discussed.
2- The methodology needs appropriate and additional references and the plant used in the biosynthesis of the AgNPs must be evidenced including by a phytochemical analysis by HPLC to be provided. Also, a voucher number could be provided. 

3- please go through the entire manuscript to correct any typos and grammatical mistakes

4- I suggest to refer and cite the following references :

a) Mohamed JMM et al. Superfast Synthesis of Stabilized Silver Nanoparticles Using Aqueous Allium sativum (Garlic) Extract and Isoniazid Hydrazide Conjugates: Molecular Docking and In-Vitro Characterizations. Molecules. 2021 Dec 24;27(1):110. doi: 10.3390/molecules27010110. PMID: 35011342; PMCID: PMC8746848.

b) Ul Haq MN et al. Green Silver Nanoparticles Synthesized from Taverniera couneifolia Elicits Effective Anti-Diabetic Effect in Alloxan-Induced Diabetic Wistar Rats. Nanomaterials (Basel). 2022 Mar 22;12(7):1035. doi: 10.3390/nano12071035. PMID: 35407153; PMCID: PMC9000644.

Best wishes,

The reviewer

Reviewer 3 Report

The manuscript is now more accurate. Introduction, results and discussion are clear and well presented. I would like to suggest some minor points to improve the reading:

In my opinion, the term Se-AgNP is confusing. It seems that NP are composed by selenium and silver. I suggest to eliminate Se or change to se-AgNP.

Line 158. Simarllery?

Line 159- Ag-nitrate :  AgNO3 or silver nitrate (chemical name)

Line 167 Silver nano-materials. ? the nanoparticle itself is not a nanomaterial

Line 170 Nine various peaks were obtained for different ratios. Details: diferent intensity ? different wavelength? One peak for each other?

Line 170. Maximum absorbance was observed at 437nm and pH 11 for 6:4 (v/v) ratios where its pointed peak indicating the formation of spherical shape (Se-AgNPs) silver nanoparticles (Figure 2).Include reference

Line 193. The crystalline nature of sliver nanoparticles was confirmed with the help of XRD analysis. Sentence in duplicate

Line 213 I suggest antibacterial activity graph instead of antibacterial  graph